# Co-Targeting Nucleus Accumbens Associate 1 and NF-κB Signaling Synergistically Inhibits Melanoma Growth

**DOI:** 10.3390/biomedicines11082221

**Published:** 2023-08-08

**Authors:** Lixiang Gu, Xingcong Ren, Chrispus Ngule, Xiaofang Xiong, Jianxun Song, Zhiguo Li, Jin-Ming Yang

**Affiliations:** 1Department of Toxicology and Cancer Biology, College of Medicine, University of Kentucky, Lexington, KY 40536, USA; lgu230@uky.edu (L.G.); xingcong.ren@uky.edu (X.R.); cmng222@uky.edu (C.N.); jyang@uky.edu (J.-M.Y.); 2Markey Cancer Center, College of Medicine, University of Kentucky, Lexington, KY 40536, USA; 3Department of Microbial Pathogenesis and Immunology, Texas A&M University Health Science Center, Bryan, TX 77807, USA; xxiong@tamu.edu

**Keywords:** NAC1, melanoma

## Abstract

Nucleus-accumbens-associated protein-1 (NAC1) is a cancer-related transcriptional factor encoded by the *NACC1* gene, which is amplified and overexpressed in various human cancers and has been appreciated as one of the top potential cancer driver genes. NAC1 has therefore been explored as a potential therapeutic target for managing malignant tumors. Here, we show that NAC1 is a negative regulator of NF-κB signaling, and NAC1 depletion enhances the level of the nuclear NF-κB in human melanoma. Furthermore, the inhibition of NF-κB signaling significantly potentiates the antineoplastic activity of the NAC1 inhibition in both the cultured melanoma cells and xenograft tumors. This study identifies a novel NAC1-NF-κB signaling axis in melanoma, offering a promising new therapeutic option to treat melanoma.

## 1. Introduction

Melanoma, an extremely aggressive form of malignancy originating from melanocytes, represents a deadly disease accounting for 80% of skin cancer deaths [1]. When melanoma is diagnosed in its early stages, the surgical resection of the lesion is associated with a favorable prognosis; however, in the advanced and metastatic stages, melanoma is generally unresponsive to various therapeutic interventions and is associated with a poor prognosis [2]. Current treatment options for advanced and metastatic melanoma have not been able to achieve greater than a 25% response rate [3]. Therefore, novel and effective therapeutic strategies for patients with melanoma are urgently needed.

Nucleus-accumbens-associated protein-1 (NAC1), encoded by the *NACC1* gene, is a transcriptional co-factor belonging to the Bric-a-Brac Tramtrack Broad complex (BTB) gene family [4]. The biologic role of NAC1 in cancer has recently emerged and this transcriptional co-factor has been found to be overexpressed in several types of cancers such as melanoma, urethral, ovarian, and lung cancer [5,6,7,8]. We previously reported that NAC1 disables cellular senescence, facilitates oxidative stress resistance during cancer progression, promotes a pro-survival autophagy through the HMGB1-mediated pathway, protects ovarian cancer from docetaxel treatment, and regulates glycolysis in ovarian cancer through its stabilization of HIF-1α [9,10,11,12,13]. More recently, we showed that NAC1 restrains antitumor immunity through the LDHA-mediated immune evasion [4]. These studies indicate that NAC1 overexpression not only leads to oncogenic transformation but may also contribute to therapeutic resistance. Nevertheless, the precise functions of NAC1 in regulating the development and progression of cancer remain incompletely understood. In this study, we uncovered a new molecular mechanism by which NAC1 regulates NF-κB signaling, and demonstrate that the combinatorial inhibition of NAC1 and the NF-κB pathway can synergistically inhibit melanoma both in cultured cells and xenograft tumors. Thus, the co-targeting of NAC1 and NF-κB may be exploited as a new effective approach to the treatment of metastatic melanoma.

## 2. Materials and Methods

### 2.1. Chemicals

BAY 11-7082 (an irreversible inhibitor of IκBα phosphorylation) and BMS-345541 (a specific IKK inhibitor) were purchased from Selleckchem (Houston, TX, USA) and dissolved in dimethyl sulfoxide as a working solution.

### 2.2. Cell Culture

SK28 cells were grown in humidified air (37 °C, 5% CO_2_) in a RPMI-1640 medium supplemented with 10% fetal bovine serum and 1% penicillin–streptomycin antibiotics. A375, A2058, and B16 were cultured in humidified air (37 °C, 5% CO_2_) in a Dulbecco’s Modified Eagle Medium (DMEM) supplemented with 10% fetal bovine serum and 1% penicillin–streptomycin antibiotics.

### 2.3. Antibodies

Antibodies against NF-κB P65 (#8242) and Lamin A/C (#4777) were purchased from CST, whereas an antibody against NAC1 was obtained from abcam (ab29047). An antibody against tubulin was purchased from Santa Cruz (sc-5286, Santa Cruz, CA, USA).

### 2.4. Western Blotting

Cells were washed with PBS after harvesting and then lysed in 20 mM of Tris (pH 8.0), 150 mM of NaCl, 1.5 mM of EDTA, 5 mM of EGTA, 0.5% Nonidet P-40, and 0.5 mM of Na_3_VO_4_ supplemented with protease inhibitors (Sigma-Aldrich, Cat#: P8340, Darmstadt, Germany). After sonication, cell lysates were collected, and protein concentrations were measured by using a Protein Assay Dye Reagent from Bio-Rad. Mix the proteins from each group with SDS-PAGE loading, respectively, and boil them for 5 min. Upon transferring to polyvinylidene difluoride membranes, proteins were probed with the indicated antibodies.

### 2.5. Depletion of NAC1

The NAC1 shRNA (short hairpin RNA) construct was transfected into the cells with a Lipofectamine 2000 reagent (Invitrogen, Carlsbad, CA, USA). Puromycin (Clontech, Beijing, China) was used to select single positive clones after transfection. After a 1-month selection, monoclones were picked up, and NAC1-deleted stable cell lines were generated.

### 2.6. Immunofluorescence Staining

For IF, cells were grown on coverslips under the culture conditions described above, fixed in a PHEM buffer with 4% formaldehyde, and blocked in phosphate-buffered saline with 5% bovine serum albumin and 0.1% Triton X-100 for 1 h. Samples were then incubated with primary antibodies against *NF-κB*, followed by incubation with secondary antibodies and DAPI.

### 2.7. RNA Isolation and Quantitative Real-Time PCR

Total RNA was extracted using an RNasy Mini Kit (Qiagen, Cat#: 74104, Germantown, MD, USA) according to the manufacturer’s instructions. The extracted mRNA was subjected to reverse transcription using a QuantiTect Reverse Transcription Kit (Qiagen, Germantown, MD, USA), following the manufacturer’s protocol. FastStart Universal SYBR Green Master was used to measure the expression level of the indicated mRNA and was normalized to β-actin, respectively. The PCR program for qRT-PCR is 95 °C for 10 min, and then repeat 40 cycles at 95 °C for 15 s and 60 °C for 30 s. The primers used in the qRT-PCR are listed as follows. Actin forward (5′CACCATTGGCAATGAGCGGTTC3′), reverse (5′AGGTCTTTGCGGATGTCCACGT3′); IL-1β forward (5′GAAATGCCACCGGGGGACAGTG3′), reverse (5′TGGATGCTCTCATCAGGACAG3′); NAC1 forward (5′CGGCTGAACTTATCAACCAGATTG3′), reverse (5′ TGACGTGGCAGTTCATCAGCTG3′).

### 2.8. Colony Formation Assay

Cells (500–1000/well) were seeded in 6-well plates and cultured in a medium for 10 days, with a medium change every 2 days. After culturing, cells were fixed in 10% formalin and stained with 0.5% crystal violet for 30 min, followed by the counting of colony numbers.

### 2.9. Mouse Xenograft Model

Mice, housed in an animal facility with free access to standard rodent chow and water, were under pathogen-free conditions and maintained in a 12 h light/12 h dark cycle. To generate tumors, SK28 cells (3 × 10^6^ cells per mouse) were mixed with an equal volume of Matrigel (Collaborative Biomedical Products, Bedford, MA, USA) and inoculated into the right flank of NSG mice (NOD scid gamma mouse). Two weeks later, animals were randomized into treatment and control groups with five mice each. Tumor volumes were measured with the following formula: V = L × W2/2 (V is volume (mm^3^), L is length (mm), W is width (mm)).

### 2.10. Statistical Analysis

The statistical significance of the results was analyzed using an unpaired Student *t* test (StatView I, Abacus Concepts Inc., Piscataway, NJ, USA). A *p* value of less than 0.05 indicates statistical significance.

## 3. Results

### 3.1. Inhibition of NAC1 Activates NF-κB Signaling in Melanoma

As the overexpression of NAC1 is involved in melanoma tumorigenesis and progression [14] and the NF-κB pathway is one of the major pathways activated in melanoma [15], we queried whether there is a functional association between NAC1 and NF-κB signaling in the context of melanoma. To test if the modulation of NAC1 could significantly alter the activity of the NF-κB pathway, B16 mouse melanoma cells were subjected to vector-based RNAi to deplete NAC1, and the siRNA-mediated down-regulation of NAC1 was examined using a quantitative real-time PCR and Western blot (Figure 1A). Notably, B16 cells with knockdown of NAC1 showed an elevation in the level of IL-1β (Figure 1D), which is a major downstream target of NF-κB [16]. To verify this observation, we investigated the NAC1-IL-1β connection in two melanoma cell lines with different genetic backgrounds: the A2058 cell line was derived from genetically engineered mouse models (Pten^f/f^) of melanoma (Figure 1B) and the SK28 cell line was established from patient-derived tumor samples and expressed mutant B-Raf (V600E) (Figure 1C). Consistently, the depletion of NAC1 in both of the cell lines markedly increased their level of IL-1β (Figure 1E,F), supporting the role of NAC1 in inhibiting the IL-1β signaling pathway. 

IL-1β is a direct target of NF-κB and contains NF-κB-binding sites in its promoter region (16); thus, we next determined whether IL-1β expression is induced by NF-κB activation. We treated SK28, A2058, and B16 cells with BMS-345541 or BAY 11-7082, the selective inhibitors of the NF-κB pathway, and RT-PCR analyses showed that NAC1-knockdown-induced IL-1β expression could be largely suppressed by the NF-κB pathway inhibitor BMS-345541 or BAY 11-7082 (Figure 1D–F). These results suggest that NAC1 knockdown results in the activation of the NF-κB signaling in melanoma.

### 3.2. Knockdown of NAC1 Leads to NF-κB Nuclear Translocation

As the active NF-κB shuttles into the nucleus to function as a transcription factor, we thus analyzed its subcellular localization in the cells subjected to the inhibition of NAC1. Nuclear and cytoplasmic extracts were prepared from SK28, B16, and A2058 cells with or without the silencing of NAC1 expression, and the level of NF-κB protein was determined with a Western blot. Figure 2 shows that all the cell lines exhibited increased levels of nuclear NF-κB following NAC1 knockdown, and treatment with TNF-α further elevated the levels of NF-κB protein in NAC1-knockdown cells and the nuclear protein Lamin A/C was not affected (Figure 2). Because the phosphorylation of Ser537 of p65 promotes its nuclear translocation and facilitates p65 binding to the promoter sequence, we proceeded to investigate whether the inhibition of NAC1 would increase the phosphorylation level of p65. We determined the level of phospho-p65 through a Western blot analysis. As depicted in Figure 2, all the cell lines demonstrated elevated levels of phospho-p65 following NAC1 knockdown. 

To validate this observation, we utilized immunofluorescence staining and confocal microscopy to detect NF-κB in SK28 cancer cells transfected with a control non-targeting RNA or NAC1-targeted siRNAs. At different time points following TNF-α treatment, SK28 cells were immediately fixed, permeabilized, and stained with antibodies. The focal location of the cell nucleus was defined using DAPI staining, and the nuclear NF-κB fluorescence was measured with an antibody against NF-κB. The rapid translocation of NF-κB from the cytoplasm to the nucleus occurred when the NAC1-knockdown cells were incubated for 10 min or 15 min with TNF-α (Figure 3). These results demonstrate that NAC1 acts as a negative regulator of the NF-κB pathway and that depletion of NAC1 induces the nuclear translocation of NF-κB in melanoma cells. 

### 3.3. Knockdown of NAC1 and Inhibition of NF-κB Act Synergistically in a Melanoma Xenograft Model

To delineate the tumorigenic roles of NAC1 in melanoma, we conditionally expressed a short hairpin RNA (shRNA) in SK28 cells, using an inducible shRNA vector to model the cellular consequence of reduced NAC1 expression. NAC1 expression was significantly knocked down in the presence of doxycycline (dox) in shNAC1 SK28 cells as compared to the WT cells, as measured with a Western blot (Figure 4A). A clonogenic assay showed that NAC1 knockdown resulted in a reduction in the colony number compared to the control (Figure 4B). Notably, the combination of NAC1 knockdown with the NF-κB pathway inhibitor BAY 11-7082 exerted a significantly stronger inhibitory effect on colony formation than the NAC1 knockdown or BAY 11-7082 alone (Figure 4B).

To recapitulate the above observation, we next tested the effect of this combinational treatment using an SK28-derived melanoma xenograft mouse model. NSG mice were inoculated subcutaneously (s.c.) with the SK28 cells stably expressing NAC1 shRNA. When tumors were formed and palpable, the mice were randomly divided into treatment groups. Mice bearing inducible-shRNA-xenografted tumors were given 2 mg/mL of doxycycline plus 5% sucrose, and monitored for tumor progression. The shCtrl SK28 cells followed the expected growth kinetics, which continued to grow until the point where ethical considerations determined the termination of the mice; knockdown of NAC1 or treatment with BAY 11-7082 alone only showed a modest inhibitory effect on tumor growth; however, the depletion of NAC1 in combination with BAY 11-7082 elicited a strong inhibitory effect on tumor growth (Figure 4C), suggesting a synergistic antitumor action between the co-targeting of NAC1 and NF-κB.

## 4. Discussion

The use of targeted and immune therapies is a major breakthrough in the treatment of melanoma patients; nevertheless, therapeutic resistance often occurs, causing failures of these treatments [17]. Therefore, melanoma remains an incurable disease, reflecting the urgency to develop new therapeutic approaches. A number of studies support the notion that NAC1 could be explored as a new therapeutic target. First, NAC1 contributes to cortactin deacetylation and augments the migration of melanoma cells [14]. Second, the expression of NAC1 contributes to immune evasion through its regulatory role in LDHA expression and lactic production in melanoma [4]. Third, NAC1 can promote glycolysis through its interaction with HIF-1a, and is critically required for the development, survival, and function of tumor cells [13]. Surprisingly, we found that one major side effect of inhibiting NAC1 is the activation of the NF-κB pathway. The inhibition of the NAC1-associated activation of NF-κB signaling is not affected by the different genetic backgrounds of melanoma cells, including the mutational status of PTEN, B-Raf (V600E), and N-Ras (Figure 1). This is of high clinical significance because melanoma is an extremely heterogeneous disease, and the identification of the conserved mechanism(s) across different subtypes of melanoma may help develop more effective drugs/approaches against this malignancy. Previous studies also linked the NF-κB pathway to melanoma tumorigenesis [18] and the activation of NF-κB has been proposed as an event that promotes melanoma tumor progression [19]. However, how the NF-κB pathway can be targeted in melanoma treatment remains unclear. Because NAC1 is involved in many melanoma-treatment-related events, we propose that the combination of NAC1 inhibition and the inhibition of the NF-κB pathway may be exploited as a novel approach for the treatment of melanoma. Recently, our lab successfully developed a novel NAC1 inhibitor [20]. Combining this inhibitor with known NF-κB inhibitors like BAY 11-7082 and BMS-345541, which target distinct steps in the NF-κB signaling pathway, may demonstrate synergistic effects. Importantly, our findings reported here suggest that the activation of the NF-κB signaling pathway needs to be carefully considered when NAC1 inhibition is used in various therapies and that targeting the inhibition of NAC1 and the NF-κB pathway simultaneously would most likely be an improved and more effective approach for melanoma therapy.

## Figures and Tables

**Figure 1 biomedicines-11-02221-f001:**
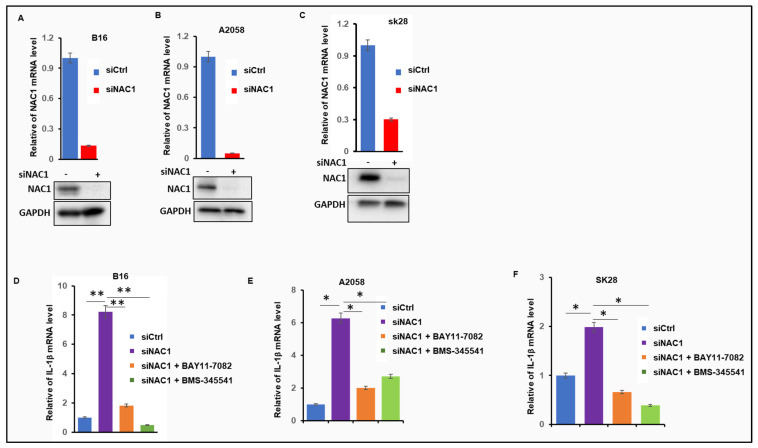
Inhibition of NAC1 activates NF-κB signaling. (**A**–**C**) Analysis of NAC1 mRNA and protein expression. Expression of mRNA or protein was assessed using RT-qPCR or Western blot at 48 h after transfection. (**D**–**F**) Effects of BAY 11-7082 or BMS-345541 on the expression of IL-1β in NAC1-knockdown SK28, A2058, and B16 cells. To investigate the influence of NF-κB pathway on the expression of IL-1β in NAC1-knockdown cells, the cells were transfected with an NAC1-specific siRNA or a non-silencing control siRNA, then treated with BAY 11-7082 or BMS-345541. Gene expression of IL-1β was measured with real-time PCR. Error bars represent SD. * *p* < 0.05 and ** *p* < 0.01.

**Figure 2 biomedicines-11-02221-f002:**
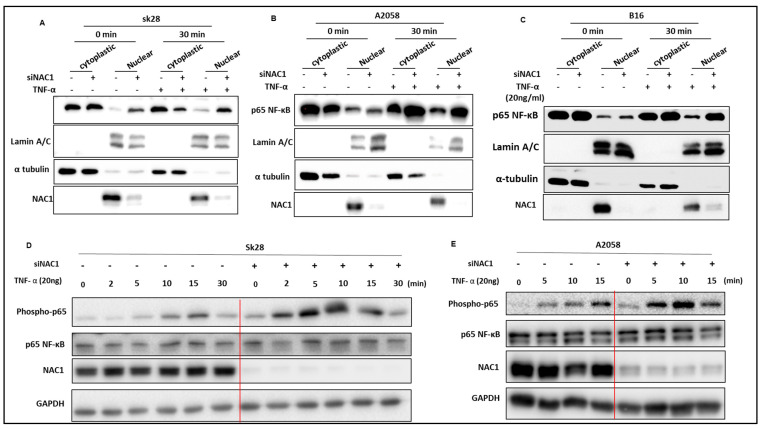
Western blotting analysis for the nuclear localization of NF-κB in melanoma cells. (**A**–**C**) Western blot analyses for the expressions of nuclear active NF-κB in SK28 (**A**), A2058 (**B**), and B16 (**C**) cells treated with either siNAC1 or TNF-α. α-Tubulin was used as a cytoplasmic marker, Lamin A/C was used as a nuclear marker. (**D**,**E**) Western blot analyses were performed to assess the levels of phosphorylated NF-κB in SK28 (**D**) and A2058 (**E**) cells treated with either siNAC1 or TNF-α.

**Figure 3 biomedicines-11-02221-f003:**
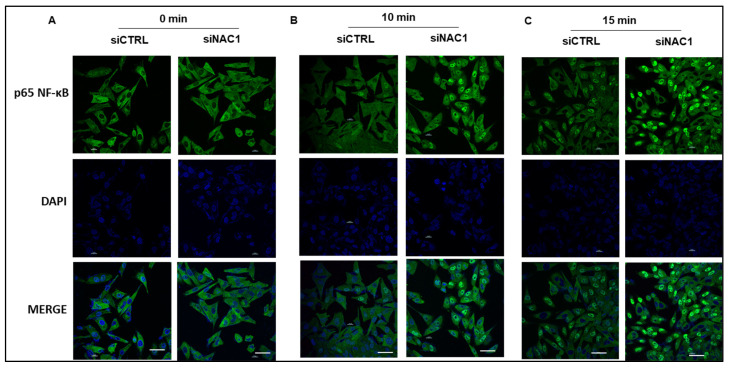
(**A**–**C**) Immunofluorescent analysis of NF-κB nuclear translocation. NF-κB (green) in siCtrl and siNAC1 SK28 cells either left untreated or treated with 20 ng/mL of TNF-α. The cells were fixed with 4% paraformaldehyde for 15 min at room temperature, permeabilized with 0.1% Triton X-100 for 15 min, and blocked with 3% BSA for 30 min at room temperature. Cells were stained with an NF-κB monoclonal antibody at a concentration of 5 µg/mL in blocking buffer for 1 h at room temperature, and then incubated with a Secondary Antibody and Alexa Fluor Plus 488 conjugates at a dilution of 1:500 for at least 30 min at room temperature in the dark (green). Nuclei (blue) were stained with Hoechst 33342. Scale bar: 50 µm.

**Figure 4 biomedicines-11-02221-f004:**
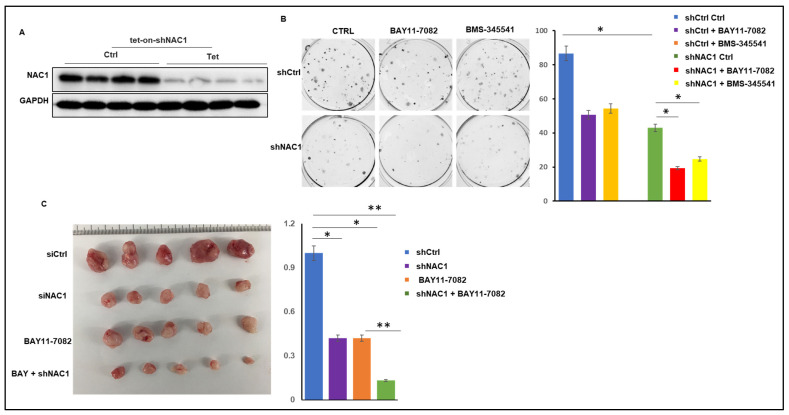
Knockdown of NAC1 increases the sensitivity of SK28 cells to BAY 11-7082. (**A**) Western blot analysis of NAC1 in SK28 cells that were stably transduced with tet-on-shNAC1. Cells were either left untreated or treated with doxycycline (100 ng/mL) for the indicated 48 h. (**B**) shCtrl or shNAC1 cells were seeded in 6-well plates and were treated with doxycycline (100 ng/mL) to knockdown NAC1 in the presence of BAY 11-7082 or BMS-345541. Upon harvesting, cells were fixed and stained with crystal violet. (**C**) SK28 cells were stably infected with lentiviral control shRNA (shCtrl) and shRNA vector-targeting NAC1 (shNAC1) and introduced to the right thigh of 6-week-old NSG mice. When the tumors of stably transfected cells reached 100 mm^3^, mice were given 5% sucrose alone or supplemented with 1 mg/mL Dox to induce NAC1 knockdown; then, the mice were treated with BAY 11-7082, and tumor size was measured twice a week. Tumor volume is presented as mean AE SEM. The photo is a representative image of the xenograft tumors. Error bars represent SD. * *p* < 0.05 and ** *p* < 0.01.

## Data Availability

All new data were provided in this manuscript.

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
