# Peer review of "Co-Targeting Nucleus Accumbens Associate 1 and NF-κB Signaling Synergistically Inhibits Melanoma Growth"

_biomedicines, 2023, doi:10.3390/biomedicines11082221_

Round 1

Reviewer 1 Report

The manuscript entitled “Co-Targeting NAC1 and NF-κB signaling synergistically inhibits melanoma growth” proposes an interesting approach, although the procedure to get that is not elucidated, as NAC1 is a negative regulator of NF-κB signaling, and the inhibition of NAC1 expression induces NFkB. This is even recognized by the authors at line 219, without any further discussion. Some possibilities, even from the speculative way, would be welcome to reinforce the manuscript. Anyway, I congratulate the authors for the conciseness in other parts of the study. In other words, is there any possible way to inhibit NAC1expresion and NFkB translocation and nuclear signal?. 

The other important concerning points is related to lines 38-40: How is that? inhibition of NFkB does not facilitate oxidative stress resistance. NFkB induces prooxidant responses. Resistance is mediated by Nrf2. Could you check and clarify this comment ?. To me, the involvement of the prooxidant response in melanoma cancer progression is unclear. Oxidative stress is a double-edge sword, and it can lead to autophagy and cell death under certain circumstances.

Minor points

 Line 53: The action of the drugs BAY11-7082 and BMS-345541 should be defined the first time they are mentioned. This action is defined at lines 129-130, but please do it before. On the other hand, use the complete definition throughout the manuscript, as they are mentioned in an abbreviated form at the legend of figure 3, lines 193 and 197.

Line 68: could you indicate the concentration and nature or sigma reference of the protease inhibitors?

Line 74: define shRNA as short hairpin RNA.

Line 103: NAC1 please.

About SK28 cell line, use always capital letters please.

Line 177: Define NSG mice.

Ref 8 and 9 are identical. One of them should be suppressed.

 Ref, 14 on, it is convenient the following the format for references. Full names of authors are not necessary. family names are enough.

Author Response

Point 1:  The manuscript entitled “Co-Targeting NAC1 and NF-κB signaling synergistically inhibits melanoma growth” proposes an interesting approach, although the procedure to get that is not elucidated, as NAC1 is a negative regulator of NF-κB signaling, and the inhibition of NAC1 expression induces NFkB. This is even recognized by the authors at line 219, without any further discussion. Some possibilities, even from the speculative way, would be welcome to reinforce the manuscript. Anyway, I congratulate the authors for the conciseness in other parts of the study. In other words, is there any possible way to inhibit NAC1expresion and NFkB translocation and nuclear signal? 

Response 1: Thank you for your valuble suggestions on our manuscript. We acknowledge that inhibiting NAC1 expression and NF-κB translocation and nuclear signaling are complex processes with potential regulatory interplay. One potential approach, as added in the discussion (lines 241-244), is to combine NAC1 and NF-κB inhibitors and test their synergistic effects in inhibiting melanoma growth and progression.

Point 2: The other important concerning points is related to lines 38-40: How is that? inhibition of NFkB does not facilitate oxidative stress resistance. NFkB induces prooxidant responses. Resistance is mediated by Nrf2. Could you check and clarify this comment?. To me, the involvement of the prooxidant response in melanoma cancer progression is unclear. Oxidative stress is a double-edge sword, and it can lead to autophagy and cell death under certain circumstances.

Response 2: Thank you for bringing up the important points related to lines 38-40 in our manuscript. Here we introudce our previously report that “NAC1 facilitates oxidative stress resistance during cancer progression”. We sincerely apologize for not clearly describing the background in our manuscript.

Point 3: Line 53: The action of the drugs BAY11-7082 and BMS-345541 should be defined the first time they are mentioned. This action is defined at lines 129-130, but please do it before. On the other hand, use the complete definition throughout the manuscript, as they are mentioned in an abbreviated form at the legend of figure 3, lines 193 and 197.

Response 3: As suggested, the action of the drugs BAY11-7082 and BMS-345541 is defined the first time they are mentioned at lines 54-55.

Point 4: Line 68: could you indicate the concentration and nature or sigma reference of the protease inhibitors?

Response 4: As suggested, Sigma reference is indicated for protease inhibitor (Sigma-Aldrich, Cat#: P8340) at line 70.

Point 5: Line 74: define shRNA as short hairpin RNA.

 Response 5: We defined shRNA as short hairpin RNA in line 76

Point 6: Line 103: NAC1 please.

 Response 6: Thak you so much, We changed NAC to NAC1 in line 116.

Point 7: About SK28 cell line, use always capital letters please

Response 7: We use capital letters SK28 in whole manuscript.

Point 8:  Line 177: Define NSG mice.

Response 8: We defined NSG as NOD scid gamma mouse in line 108

Point 9: Ref 8 and 9 are identical. One of them should be suppressed.

Done

Response 9: Thank you so much, it should be Ref 9 and 10 identical, we correct Ref 9.

Point 9: Ref, 14 on, it is convenient the following the format for references. Full names of authors are not necessary. family names are enough.

Response 9: Thank you so much for your suggestions, we followed the format for references.

Reviewer 2 Report

A study by Gu et al investigates co-Targeting NAC1 and NF-κB signaling to synergistically inhibit melanoma growth. The topic of the study is novel and interesting. The communication is well written and conclusions are generally supported by the results.

Specific comments:

1. It is not clear why NF-kB is written in italics.

2. The Authors must clearly state what they mean for "NF-kB". The antibodies they use (#8242) recognize p65 subunit irrespectively of its activation status. The authors should change labels in figures e.g., p65/NF-kB?

3. Please specify types of mice used in the study.

4. M&M section lacks description of the analysis of mRNA levels.

5. Fig. 2 would benefit from immunoblotting of phosphorylated p65.

6. Fig. 3 - scale bars are missing.

Author Response

Point 1:  A study by Gu et al investigates co-Targeting NAC1 and NF-κB signaling to synergistically inhibit melanoma growth. The topic of the study is novel and interesting. The communication is well written and conclusions are generally supported by the results.

Response 1: Thank you for your positive feedback on our study. We are grateful for your recognition of the novelty and interest of our research topic.

Point 2:  It is not clear why NF-kB is written in italics.

Response 2: NF-κB should not be written in italics. We have rectified the issue in our manuscript. NF-κB is a common abbreviation for the nuclear factor kappa B, and it should be written in regular font without italics.

Point 3:  The Authors must clearly state what they mean for "NF-kB". The antibodies they use (#8242) recognize p65 subunit irrespectively of its activation status. The authors should change labels in figures e.g., p65/NF-kB?

Response 3: Thank you for bringing up the important point regarding the clarification of "NF-κB" in our manuscript. We changed the labels in Fig. 2 and Fig 3A accordingly.

Point 4: Please specify types of mice used in the study.

Response 4: NSG mice (NOD scid gamma mouse) were used in the study.

Point 5: M&M section lacks description of the analysis of mRNA levels.

Response 5: The Materials and Methods (M&M) section now includes a detailed description of the analysis of mRNA levels (2.7. RNA isolation and quantitative real-time PCR).

Point 6: Fig. 2 would benefit from immunoblotting of phosphorylated p65.

Response 6: Thank you for your suggestion. We have incorporated the testing of phosphorylated p65 and included the results in Fig. 2.

Point 7: Fig. 3 - scale bars are missing.

Response 7: Scale bars have been included in Fig. 3.

Reviewer 3 Report

This is an important and promising communication concerning new therapies of melanoma, however, there are somer doubts to be considered or improved before the paper becomes ready for publication.

1. It could be expected to compare the used lines of melanoma with uveal melanoma, if possible, as they sometimes are regulated completely differenlly

2. Isn't it possible to compare the experiments with normal (not transformed) melanocytes? It would share some more light on the observations

3. Is it possible to evaluate whether melanin production in the observed lines is affected by the studied NAC1 expression? It may strongly influence the studied therapeutic effects.

4. As you yourselves mentioned about the extremal heterogeneity in melanoma lines, please justify the normal distribution assumption for the justification of the Student's t test usage, otherwise a non-parametric text should be employed.

5. I cannot find whether the error bars of the figures represent SD (as it should be) or SEM which represents a completely different interpretation, not related to the population of observations.

6. There are numerous errors of English grammar and spelling. Please maintain consequently Western blot or western blot, SK28 or sk28 (if it is the case), please use a space between number and unit and keep it uniformly. Please double check spelling and editorial format of the text.

English is properly followed, but there are some errors mentioned above.

Round 2

Reviewer 3 Report

The manuscript has gained un quality and is now good enough for publication